# Disruptions to the procurement of medical abortion medicines during COVID-19: a scoping review

Natasha Cassinath,[1] Patricia Titulaer,[1] Laurence Läser,[1] Antonella Lavelanet,[1] Safia Ahsan,[2] Francelle Kwankam Toedtli,[3] Stephen Mawa,[3] Ulrika Rehnstrom Loi  [1]

¹UNDP-UNFPA-UNICEF-WHO-World Bank Special Programme of Research, Development and Research Training in Human Reproduction (HRP), Department of Sexual and Reproductive Health and Research, World Health Organization, 20 Avenue Appia, 1211 Geneva, Switzerland
²Reproductive Health Supplies Coalition, Washington, DC, USA
³Technical Division, United Nations Population Fund, New York, New York, USA

**Correspondence to**
Dr Ulrika Rehnstrom Loi;
rehnstromu@who.int

## ABSTRACT

**Objectives** This scoping review aimed to systematically search, retrieve and map the extent and characteristics of available literature on the evidenced disruptions to medical abortion (MA) medicine procurement caused by the COVID-19 outbreak.

**Design** Scoping review using Arksey and O'Malley's methodology and Levac *et al*'s methodological enhancement with adherence to the Preferred Reporting Items for Systematic Reviews and Meta-Analyses extension for scoping reviews.

**Data sources** PubMed, Embase, PMC, Science Direct, the Cochrane Library and Google Scholar were searched from January 2020 to April 2022.

**Eligibility criteria** We included articles in English that: (1) contained information on MA medicines; (2) included descriptions of procurement disruptions, including those with examples, characteristics and/or statistics; (3) documented events during the COVID-19 pandemic; and (4) presented primary data.

**Data extraction and synthesis** Two reviewers independently screened search results, performed a full-text review of preliminarily included articles and completed data extraction in a standard Excel spreadsheet. Extracted data from was compared for validation and synthesised qualitatively.

**Results** The two articles included are unpublished grey literature demonstrating evidence of short-lived disruptions in sexual and reproductive health commodity procurement, including MA medicines, in sub-Saharan Africa during the early months of the pandemic. Findings from the two included grey literature articles show that in sub-Saharan contexts, emergency preparedness, stockpiling, adaptations and flexibility of key actors, including donors, alleviated COVID-19 disruptions allowing for resumption of services within weeks.

**Conclusion** There is a need for increased empirical evidence of MA procurement challenges to understand which barriers to MA procurement may persist and impact continuity of supply while others can fuel resilience and preparedness efforts at the country and subregional levels. The lack of evidence from social marketing organisations and their networks is a significant gap as these actors constitute a vital artery in the distribution of MA commodities in low-income and middle-income countries.

### STRENGTHS AND LIMITATIONS OF THIS STUDY

⇒ We considered 1662 articles related to 'medical abortion' and 'COVID-19'.
⇒ The study uncovered critical areas for future medical abortion supply chain research.
⇒ The available evidence for extraction was limited to two articles that were not published in peer-reviewed journals.
⇒ The articles reviewed address supply chain issues of sub-Saharan African countries with only a brief discussion of procurement in each article and unclear distinction of challenges by type of procurer.
⇒ As studies refer to sexual and reproductive health including medical abortion, there is a risk that some interventions or conclusions are erroneously attributed to the challenge of medical abortion procurement.

## INTRODUCTION

Globally, roughly 73.3 million abortions take place per year, at least half of which are performed through medical abortion (MA).[1 2] MA is a procedure that terminates pregnancy using medicines—either mifepristone and misoprostol or misoprostol alone.[3] Mifepristone is a powerful abortifacient that blocks progesterone receptors and glucocorticoid receptors in the placenta to stop growth. Misoprostol dilates the cervix and induces muscle contractions that clears the uterus. MA is widely considered to be a safe and effective means to terminate pregnancy.

MA medicines advanced the ability of women to obtain abortion safely, at home, and with limited or no interactions with healthcare providers, if desired. MA medicines figure on the WHO essential medicines list (EML) and are considered vital for comprehensive and postabortion care.[4 5] Country-based availability of MA medicines is restricted to where there is at least one legal ground for abortion. In 2019, 129 countries of a study of 158 had such a legal basis for

abortion and, in principle, market authorisation to stock and distribute MA medicines although in a controlled manner.[6]

Access to MA medicines is contingent on conducive policy, regulation, registration, national EML inclusion, financing and supply chain management. When COVID-19 was declared a public health emergency of international concern by the WHO in January 2020, the restrictions in movement of goods and people strained supply chains and put the procurement of MA medicines in jeopardy. Procurement is the act of obtaining goods and services at an appropriate level of quality, at a specified time and location and at a negotiated price.[7] It relies on established relationships with domestic and international suppliers; the capacity to accurately forecast demand; and the management of the supply pipeline to avoid stockouts, warehousing challenges or product expiry. All of these complex activities were reportedly compromised when domestic and international manufacturing, transport, donors and regulations were impacted by COVID-19 containment efforts, just as MA procurement was needed more than ever.[8]

In some settings, MA medicines obtained by mail or locally with or without health provider assistance presented women with the option to bypass lockdown or distancing restrictions, and in others, accelerated the political acceptance of self-managed abortion as it became an urgent necessity for service continuity.[9–11] When the virus containment measures started to strain global supply chains and divert funds and energy to personal protective equipment (PPE), key stakeholders—armed with little information—anticipated that the procurement of MA medicines was equally at risk and called for urgent action and preparedness, including stockpiling and close surveillance of the supply chains to offset potential procurement challenges.[12–17]

The United Nations Development Programme (UNDP) –United Nations Population Fund (UNFPA)– United Nations Children's Fund (UNICEF) – World Health Organization (WHO) –World Bank Special Programme of Research, Development and Research Training in Human Reproduction (HRP) conducted a prescoping survey to better understand the impact of COVID-19 on the ability to fund and procure MA medicines. There were 22 respondents from UNFPA, non-governmental organisations (NGOs) and social marketing organisations. This resulted in a list of supply chain funding and procurement challenges, as well as successful adaptations. However, no clear root causes of disruptions, pervasive challenges or mitigation measures were identified. The rationale for this scoping review was to explore what emerging evidence on MA medicines procurement disruptions existed over the first 2 years of the pandemic and identify research gaps. Understanding the causes of MA supply chain fragility or resilience in the short and longer term is essential to the development of mechanisms to forecast and plan for reliable supply in uncertain future scenarios. This understanding is of interest to comprehensive abortion care stakeholders, including actors involved in MA supply procurement, risk management, emergency preparedness and donors. Thus, the aim of this scoping review is to systematically search, retrieve and map the extent and characteristics of available literature on the disruptions to MA medicine procurement during the COVID-19 outbreak.

## METHODS
### Study design
We undertook a scoping review to explore the nature and extent of the evidence related to disruptions to procurement of MA medicines during COVID-19. We considered further areas of inquiry related to the types of disruption, causes and adaptations but intentionally left the question broad to be as inclusive as possible. The methods for this review were informed by Arksey and O'Malley's methodology and Levac *et al*'s methodological enhancement.[18 19] This paper adheres to the Preferred Reporting Items for Systematic Reviews and Meta-Analyses (PRISMA) extension for scoping reviews.[20] The PRISMA checklist can be found in online supplemental appendix 1.

### Patient and public involvement statement
During the planning phase, we consulted partners and stakeholders through a virtual survey and several virtual meetings to engage them in the scoping review and garner their support to make sense of the study findings.

### Search strategy
We piloted different search strategies including the concepts of 'medical abortion', 'procurement' and 'COVID-19' with no geographical limitations. One reviewer (NC) supported by four advisory team members (PT, LL, AL and URL) harvested keywords from similar studies and trialled these in PubMed. We developed the final search strategy in consultation with the WHO medical librarian. We reduced the number of concepts to two: 'medical abortion' and 'COVID-19' and included studies from January 2020 (right before the pandemic was officially declared) to April 2022 (fourth to sixth wave depending on location). The comprehensive search strategy is presented in online supplemental appendix 2. We conducted a database search in PubMed, PubMed Central, Embase, Science Direct, the Cochrane Library and the WHO COVID-19 database between November 4 and 6, 2021 and refreshed it on 12 April 2022. We searched Google Scholar with additional filters ('supply' OR 'supply chain' OR 'disruption') to manage the volume of results. Additional grey literature was obtained by searching target organisations' websites (eg, UNFPA, WHO, MSI Reproductive Choices, DKT International and Reproductive Health Supplies Coalition). The reviewers used forward and

backward citation searching of all articles in the full-text review stage to find any additional studies fitting the inclusion criteria. All search results were stored using EndNote reference manager software and uploaded to Covidence systematic review software.[21 22]

### Study selection and data extraction process

Two reviewers (NC and PT) screened all titles and abstracts for inclusion independently. Studies that appeared relevant in a first instance were subjected to a full text review. We included articles that met the following inclusion criteria: (1) contained information on MA medicines; (2) included descriptions of procurement disruptions, including those with examples, characteristics and/or statistics; (3) documented events during the COVID-19 pandemic; and (4) articles presenting primary data. These criteria were developed in consult with all team members, based on the preliminary results of the search. We considered all English language documents from the timespan referred previously. A full description of the search terms, results and inclusion/exclusion criteria is found in online supplemental appendix 2.

We developed a data extraction form in a standard Excel spreadsheet. The data points included document and study characteristics: author(s), title, publication year, journal, country/region, aim, nature of disruptions, root causes, adaptations and author's conclusions. Two reviewers independently conducted the data abstraction. Each reviewer's extraction was compiled into a single excel spreadsheet in Microsoft Excel for validation.[23]

### Data synthesis

Categorised data in the extraction table (tables 1 and 2) were analysed across studies to identify emerging themes and subthemes related to the research question. Debriefing sessions were held with two members of the team to improve rigour and reduce researcher bias. Limitations to the heterogeneity of results were acknowledged. Study quality was not assessed as this is not a requirement for scoping reviews.[18]

## REVIEW FINDINGS

Figure 1 shows the PRISMA flow chart. The search strategy returned 1662 results from the search of 6 databases, Google Scholar and web search. Of these, we removed 816 duplicate records and excluded 811 records that did not meet the inclusion criteria. We conducted a full text review of 35 articles. Ultimately, we extracted information from two documents.

### Characteristics of included articles and data extraction

Table 1 outlines the characteristics of the two studies included in the scoping review. Both articles present national level data on supply chain disruptions to sexual and reproductive health (SRH) or family planning commodities including MA medicines. Table 2 outlines the data extracted from the selected articles.

The findings from the included articles were organised into five areas related to system-level MA medicines procurement during the COVID-19 pandemic: (1) prepandemic stockout mitigation strategies and their effects during the pandemic; (2) procurement adaptations at the international and local levels made in response to the pandemic; (3) the impact of manufacturing and transport related disruptions to procurement; (4) enabling or disabling national policies put in place during the pandemic; and (5) cost increases and changes to budget management due to the pandemic.

Both articles described stockouts of SRH medicines as a regular occurrence in many countries prepandemic.[24 25]

| Table 1 | Study characteristics | | | | | | |
|---|---|---|---|---|---|---|---|
| Author(s) Year | Article type | Journal | Data collection | Region | Aim | | Commodity/ medicine discussed |
| Reproductive Health Supplies Coalition (RHSC) 2021[24] | Rapid assessment | Grey literature | 53 interviews conducted globally. Data collected March–October 2020* | Global, with additional quantitative data from Ethiopia, Zambia and Kenya. | To understand the disruptions to SRH supply chains during the pandemic and chart a roadmap for supply chain resiliency. | | General SRH including MA. |
| Otieno *et al* 2021[25] | Rapid needs assessment | Grey literature | Survey data from 17 African ministries of health/data collected July 2020 | 17 sub-Saharan African countries. | To assess and document continuity of SRH services with a focus on safe abortion, postabortion care and family planning services during the COVID-19 pandemic. | | Family planning and MA. |

*This study included prepandemic data. Prepandemic and COVID-19 specific data were distinguishable.
MA, medical abortion; SRH, sexual and reproductive health.

**Table 2** Data extraction – disruption, adaptation and author's conclusions

| Author(s) year | Types of disruption | Adaptations | Authors' conclusions |
|---|---|---|---|
| RHSC 2021[24] | Quarantine at ports/border closures; manufacturing stoppage; shortage of freight containers; increased product and freight costs; lack of air cargo/increased demand for sea freight; workforce reduction; reduced demand for MA; uncertainty due to changing restrictions; regulatory approvals; and delay in reauthorisation of products. | Diversification of delivery channel; transition to sea freight; mapping and tracking adaptations; dispensing guidelines were changed; funder flexibility and responsiveness. Policy guidance from WHO and UN agencies helped overcome bottlenecks, particularly around the movement of health products and the inclusion of SRH within essential services. | Manufacturing, logistics and systems (including policies and procedures) were most affected. Data from six countries showed no unusual changes in stock levels. Higher initial stock levels mitigated delays. Within 2–4 months, most companies and organisations resumed operations with new constraints. Many health supply chains rely on material inputs from China and India, including active pharmaceutical ingredients. Transportation will continue to be challenging. |
| Otieno *et al* 2021[25] | Limited public procurement; funding shortfall; general stock delay | None related specifically to procurement. | MA drugs available in health facilities in 13 (76%) countries. Where not available, reasons included: abortion forbidden, drugs not in health facilities but available in the pharmacies, controlled drugs not available for MA, available but for induction of labour and for postpartum haemorrhage, and abortion drugs not included on the essential medicines list. Madagascar had no abortion care due to lack of trained healthcare workers. |

MA, medical abortion; SRH, sexual and reproductive health.

In some countries, this risk was mitigated by maintaining a large stockpile of some of these medicines. During COVID-19, this protective overstock may have served to lessen the levels of stockouts in the short term. However, by the end of 2020, there was evidence of increased stockout in Zambia, Ethiopia and Kenya, where stocks had otherwise remained stable.[24] In other sub-Saharan Africa countries, COVID-19 related stockouts of SRH commodities began in March 2020 or later and continued to occur until at least October 2020. During this period, procurers were faced with increased costs related to PPE and pandemic response, commodity purchase prices and transport; limited time to negotiate or perform price comparisons; and the additional effort to secure the transport and fulfil the purchase orders. These hurdles were followed by the implementation of adaptations allowing for abortion services to return to prepandemic levels within 'weeks, not months'.[24] Adaptations included attempting to identify alternative suppliers; adjusting to limited transport options; mapping and tracking orders; lobbying for funder flexibility and responsiveness; redistributing stock; and advocating for SRH and/or MA supplies priority.[24] Diversifying the supplier base was a key challenge as manufacturers are largely geographically concentrated in Asia.[24]

At the onset of the pandemic, manufacturers stated that fluctuations in demand, anticipated supply chain disruptions and panic buying challenged their ability to plan and produce SRH supplies in step with actual demand, but they were able to rapidly return to prepandemic production levels once they adopted COVID-19 protocols in their work environment.[24]

Replenishment stock of SRH commodities through usual channels proved challenging, but in some instances, there was evidence of local coordination and collaborative management of inventory between procurers, such as service providers and retailers, allowing for healthy stocks in one region to be redeployed to another.[25] WhatsApp was used to improve communication and support the coordination of stock availability and redistribution.[24 25]

National policies related to the movement of select goods and workers played a part in stalling the ability of procurers to receive MA medicines. When abortion was not immediately deemed an essential service, the flow of MA medicines was deprioritised and prone to delay.[25] When MA was designated as an essential medicine, it positively impacted the ability to receive and distribute MA medicines nationally.[24 25]

Increases in transport costs and the diversions of funds to PPE were found to have direct impacts on SRH and MA budgets.[24 25] However, the burden of these costs was not always carried by the procurer—manufacturers, wholesalers and donors also incurred a share of these costs.[24] In the short term, donor flexibility; budgetary reallocation from PPE to SRH commodities or vice versa; and/or the ability to increase prices downstream in the supply chain allowed for continued procurement. The WHO African Region found that health ministries' financing for national procurement was stable in 2020 as funding had been allocated and orders were already placed prepandemic.[25]

Concerns related to postpandemic costs were raised in both articles.[24 25] Manufacturers claimed that improving risk mitigation efforts for the future, such as by maintaining larger stockpiles or developing relationships with a more diversified supplier base, would drive up their costs and affect product pricing.[24] Both articles highlighted that transport costs may continue to be impacted

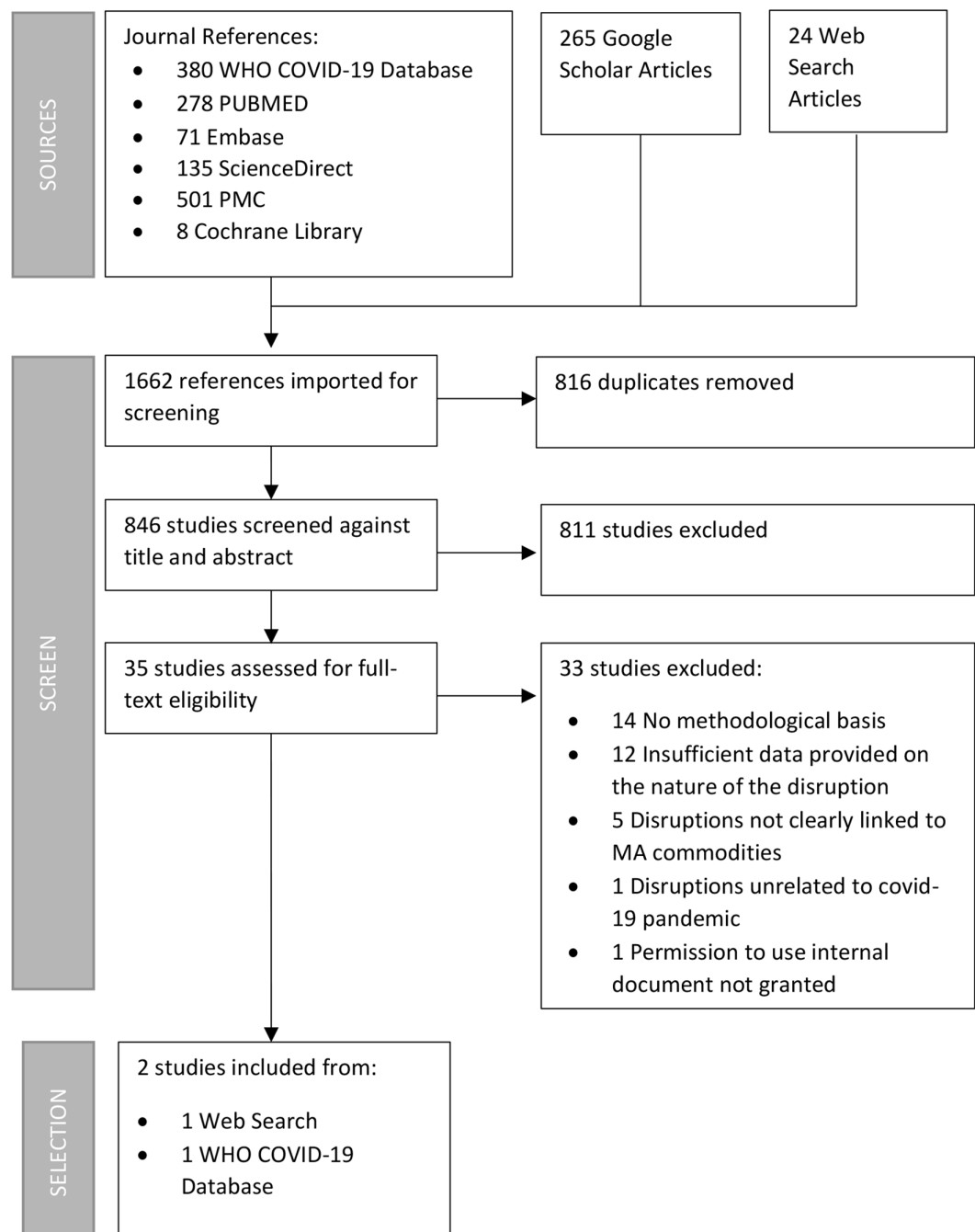

**Figure 1** PRISMA flow chart. PRISMA, Preferred Reporting Items for Systematic Reviews and Meta-Analyses.

by backlogs, travel restrictions, storage constraints and limited freight options.[24 25]

## DISCUSSION

While there exists no straightforward data regarding the prevalence of MA administration, proxy information from studies looking at the readiness of 210 health facilities in 11 African countries to provide postabortion care demonstrates that the primary challenge is not the availability of uterotonics (misoprostol or oxytocin), but rather, limitations of staff and critical care unit infrastructure.[26]

Before the COVID-19 pandemic, unintended pregnancy rates were higher in low-income and middle-income countries (93 and 66 unintended pregnancies per 1000 reproductive-aged women per year, respectively) than in high-income countries (34 per 1000).[1] Before the pandemic, women in low-income and middle-income countries were also less likely to have a safe and legal abortion. For example, the most recent data available indicate that in sub-Saharan Africa between 2015 and 2019, 8.0 million annual abortions occurred and between 2010 and 2014, 77% of abortions were unsafe.[2] Based on past experience of pandemics and other disruptive events,

the magnitude of these disparities would be expected to increase the need and challenge the access to abortion services and products.[27]

Despite the anticipation of catastrophic impacts of COVID-19 on the supply of abortion medicines, this scoping review, employing a systematic comprehensive search strategy to gather all types of available evidence, found an almost negligible number of articles (2) on the topic, both grey literature.

The limited evidence uncovered that preparedness—namely buffer stocks—and rapid adaptation to new conditions by multiple stakeholders (donors, policy makers, manufacturers and NGOs) were key ingredients in ensuring the continuous supply of SRH commodities, including MA medicines.

Concerns around COVID-19 disruptions on family planning were justified based on past experience, but ultimately, the available evidence indicates that the scale of disturbance in the select sub-Saharan countries was not as detrimental as was initially envisioned and/or service returned to prepandemic levels once lockdown periods concluded.[28 29]

It remains to be known how expansively resilience was experienced, where and how long it can be expected to hold given that continued COVID-19 risk mitigation measures, pre-emptive stockpiling, limited transport options, shipping container shortages and rising petrol prices will increase the cost of doing business. National health budgets will be stretched as COVID-19 testing and vaccination efforts continue, and funds may be redirected to general population health and mental health due to neglect over the last 2 years.[30 31] In addition, organisations or companies reliant on international donations and/or unable to absorb these increased costs are anticipated to face a tight financing landscape, particularly due to the reduced levels of funding for safe abortion worldwide.[32]

It cannot be overstated that MA commodity medicines are difficult to obtain in many parts of the world and that 'returning to normal' does not imply a satisfactory status quo, particularly in restrictive jurisdictions or fragmented markets where data sets are unavailable.

### Strengths and limitations

The strengths associated with this scoping review are that it includes both published and unpublished literature and provides an overview of current evidence in the field of MA medicine procurement caused by the COVID-19 outbreak. However, there are several limitations to this review. Our focus was strictly on MA procurement, and therefore, the availability of MA medicines does not infer access by clients. The available evidence was limited and not published in peer-reviewed journals. There is only a brief discussion of procurement in each article, in favour of a broader discussion of supply chain issues affecting either SRH supplies broadly or family planning supplies specifically. The studies refer to SRH including MA, which made applicability of some interventions or conclusions difficult to attribute to MA procurement specifically.

Importantly, the evidence from field-based NGOs and global social marketing organisations is very limited in these studies. Social marketing organisations are recognised as important actors in the procurement and distribution of MA services and products in low-income and middle-income countries yet no published documentation of their experiences with MA procurement during the COVID-19 pandemic was found. Additionally, the articles reviewed were heavily weighted towards the supply chain issues of sub-Saharan African countries and may not be relevant or representative of the reality in other geographies.

### CONCLUSION

This scoping review identified evidence gaps and future research needs to: (1) improve and increase the empirical evidence on MA medicines procurement and the greater supply chain and (2) improve the resilience and preparedness of state and private health systems distributing MA medicines. Given the general nature of current research looking at multiple countries across multiple SRH products, further examination should be undertaken in specific countries where market fragility is known to exist to support risk mitigation and preparedness efforts. The most important contribution to this work would be in capturing the resilience of social marketing organisations and their network who distribute the bulk of MA medicines in low-income and middle-income countries worldwide. These organisations have outsized influence and experience in the procurement and distribution of MA products, particularly in low-income and middle-income countries.[33]

**Acknowledgements** The authors would like to acknowledge the input of Kavita Kothari, WHO Librarian, Geneva, Switzerland, for her suggestions and support in refining the search strategy.

**Contributors** The scoping review was conceptualised by URL, PT, SA and FKT. NC and PT conducted the review, including assessing literature and data synthesis. NC prepared the first draft manuscript with significant intellectual contributions from PT, AL, LL, SA, FKT, SM and URL. The authors alone are responsible for the views expressed in this article, and they do not necessarily represent the views, decisions, or policies of the institutions with which they are affiliated. All authors that also reviewed and edited versions of the manuscript. URL is responsible for the overall content as guarantor by accepting full responsibility for the finished work and the conduct of the study. URL had access to the data, and controlled the decision to publish. All authors approved the final manuscript for publication.

**Funding** This work was supported by the UNDP–UNFPA–UNICEF–WHO–World Bank Special Programme of Research, Development and Research Training in Human Reproduction, a cosponsored programme executed by the WHO.

**Disclaimer** The views expressed in this article are those of the authors and do not necessarily represent the views of, and should not be attributed to, the WHO or the United Nations Population Fund.

**Competing interests** None declared.

**Patient and public involvement** Patients and/or the public were not involved in the design, or conduct, or reporting, or dissemination plans of this research.

**Patient consent for publication** Not applicable.

**Ethics approval** An ethics statement is not applicable because this study was based exclusively on published literature.

**Provenance and peer review** Not commissioned; externally peer reviewed.

**Data availability statement** No data are available. No additional data available.

**ORCID iD**
Ulrika Rehnstrom Loi http://orcid.org/0000-0002-3455-8606

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
