## [Reviewer comments · BMJ Open]

ARTICLE DETAILS

TITLE (PROVISIONAL)	Disruptions to the procurement of medical abortion medicines during COVID-19: a scoping review
AUTHORS	Cassinath, Natasha; TITULAER, Patricia; Läser, Laurence; Lavelaneta, Antonella; Ahsan, Safia; Toedtli, Francelle; Mawa, Stephen; Rehnstrom Loi, Ulrika

VERSION 1 – REVIEW

REVIEWER	Brhlikova, Petra Newcastle University, Population Health Sciences Institute
REVIEW RETURNED	20-Jun-2022

GENERAL COMMENTS	Thank you for the opportunity to comment on this interesting review. My comments are primarily minor and follow the organization of the material in the provided pdf. 1. Abstract – more details on results would be useful; reduce conclusion – e.g., the significance of the last sentence to the results the current review is not clear2. Box 1 – The abbreviation ‘MA’ is not introduced here. It is used only twice so I would suggest to spell out. Introduction 3. The first paragraph in Introduction needs to be supported by references, eg., WHO guidelines on “Medical management of abortion” and WHO EML.4. Some of the references mentioned at the end of the 3rd paragraph in the Introduction focus on contraception and are not directly relevant here (eg., no. 6, 8, 10).5. Introduction, 4th paragraph, ll.32-39 – is there an output from the survey which could be referenced here?6. The SMOs abbreviation is introduced here but it is used only once in the whole manuscript. I suggest to omit this one to reduce the number of abbreviations. Methods section 7. Despite the rush to publish covid-related studies, the two-year period is relatively short for academic studies to be conducted and published. Could you clarify if pre-prints were captured in the searches of academic databases? Review findings 8. 2nd sentence (p.7, about ll.23-25) -Please change the order; first remove duplicates then those not meeting inclusion criteria (in the current order, the resulting number is incorrect).
--

	Characteristics of included articles ... 9. p.9, l.27 – the second reference provided here should be no. 19 not 20 (?) Discussion 10. 2nd paragraph – supply to clients. The focus of the review is on procurement (i.e. availability) of MA medicines which does not equal to appropriate access. This should be made explicit here or in the limitations paragraph. Even if medicines were available at clinics and pharmacies, the medical abortion service provision was likely curtailed due to lock downs and priority given to covid patients. Moreover, if only relying on pharmacies and drug shops or online orders, information on effective regimens provided to women might be poor (e.g., Footman et al. 2018 ‘Medical Abortion Provision by Pharmacies and Drug Sellers in Low- and Middle-Income Countries: A Systematic Review). Strengths and limitations 11. l.11 ‘an overview’ instead of ‘on overview’ Conclusion 12. SMOs – please spell out 13. The role of social marketing organisations is highlighted (in abstract-conclusion and manuscript conclusion). The results section, however, does not make it entirely clear if the observations relate to public health systems, services provided by non-governmental organisations, or both. One mention of ministries suggests that the available evidence refers to public systems. It is important to clarify this point to provide an accurate account of what is known and what is not known. Supporting references for social marketing organisations distributing ‘the bulk of medical abortion medicines’ being distributed (and procured?) and their ‘outsized influence’ are needed.
--	--

REVIEWER	Ennis, Madeleine The University of British Columbia Faculty of Medicine
REVIEW RETURNED	19-Jul-2022

GENERAL COMMENTS	Summary: The authors conducted a scoping review to systematically map the available literature on the disruptions to medical abortion medications caused by the COVID-19 pandemic, with two unpublished articles included, demonstrating limited evidence of brief disruptions to medical abortion medications in Sub-Saharan Africa early in the pandemic. This publication makes a strong argument for needing to understand barriers to accessing medical abortion medications, and has the potential to facilitate collaboration, and allow researchers, medical staff, clinicians, and policy makers to stay up-to-date with issues in modern abortion care. It highlights that more research is needed to understand this issue, as there is a real scarcity in data. After a Pubmed search, I concluded that this is original research, with unique objectives. This being said, I recommend substantial changes to this article. Firstly, the language in the manuscript should be altered to reflect biological sex, and not gender (e.g. refer to females instead of women when making general statements). This language recognizes diversity and is more inclusive. The comment “Women in this scoping review is intended to be construed inclusively to include girls and all persons who can become pregnant.” could then be removed. There is missing punctuation and a number of typos that should be corrected. I would recommend a more in-
---

	depth description of medical abortion medications, and the pre-pandemic availability of these medications around the world, and especially in sub-Saharan Africa. Many of the conclusions and statements made throughout the article need to be revised to reflect that the available data was focused on sub-Saharan Africa and was unpublished, so as to not over state or over generalize their findings. Specific comments: Introduction:  1. Be consistent abbreviating MA. 2. The introduction should include a more information on the number of abortions accessed annually using medications. What are the currently available medications, and how does their availability differ around the world prior to the pandemic? Not just from a disruptions stand-point, but also from a regulations and legality perspective. Methods:  1. The limitation that the studies refer to SRH including MA should be made clearer in the methods. Review findings:  2. It is very unclear throughout this section which medications are being described. Mifepristone, misoprostol, methotrexate, etc. Please elaborate on this throughout. Discussion:  1. You make it clear that there is a lack of literature on this topic and more research is needed to address the issue of MA medication access. However, your discussion is too short, and does not give the readers and good understanding of the global landscape of abortion care access, including access to medications. There should be more comparison to the literature, and authors thoughts on further solutions to this problem, possibly by describing facilitators to other medications around the world. 2. The limitation that the articles found focus on sub-Saharan Africa should be discussed more thoroughly throughout the discussion, including providing more context as to how abortion care is normally accessed there. Small details:  1. Consider switching medicines to medications, and be consistent in pluralizing it throughout.
--	--

VERSION 1 – AUTHOR RESPONSE

Reviewer: 1

Dr. Petra Brhlikova, Newcastle University

Comments to the Author:

Thank you for the opportunity to comment on this interesting review. My comments are primarily minor and follow the organization of the material in the provided pdf.

1. Abstract – more details on results would be useful; reduce conclusion – e.g., the significance of the

last sentence to the results the current review is not clear

Author's comments: The conclusion was reworded for clarity and the findings were enhanced, page 2.

2. Box 1 – The abbreviation 'MA' is not introduced here. It is used only twice so I would suggest to spell out.

Author's comments: Box 1 was removed as per editorial comments.

Introduction

3. The first paragraph in Introduction needs to be supported by references, eg., WHO guidelines on "Medical management of abortion" and WHO EML.

Author's comments: Thank you for noting this omission. These references have been added, page6, line 10-16.

4. Some of the references mentioned at the end of the 3rd paragraph in the Introduction focus on contraception and are not directly relevant here (eg., no. 6, 8, 10).

Author's comments: Thank you very much for this great comment. Due to the BMJ referencing convention, references must be placed at the end of a sentence. These references refer to the first part of the sentence - the basis upon which key stakeholders assumed abortion supply chains would be as affected as other supply chains, most notably contraception. These references were taken from documents where authors extrapolated the impact on MA supply chains based on these (referenced) texts. So, here we are not substantiating the evidence of the effect of COVID-19 on the MA supply chain, but rather, we are sharing the texts upon which inferences were made by stakeholders. We have reworded the sentence for clarity Page 6.

5. Introduction, 4th paragraph, ll.32-39 – is there an output from the survey which could be referenced here?

Author's comment: This is an internal document that the WHO has not made public. Minor revisions were made on page 7.

6. The SMOs abbreviation is introduced here but it is used only once in the whole manuscript. I suggest to omit this one to reduce the number of abbreviations.

Author's comment: The abbreviation has been removed.

Methods section

7. Despite the rush to publish covid-related studies, the two-year period is relatively short for academic studies to be conducted and published. Could you clarify if pre-prints were captured in the searches of academic databases?

Author's comment: As per Appendix 2, all types of articles were included except protocols. Pre-prints with very limited details were not included because we could not establish the evidence base.

Review findings

8. 2nd sentence (p.7, about ll.23-25) -Please change the order; first remove duplicates then those not meeting inclusion criteria (in the current order, the resulting number is incorrect).

Author's comments: The order has been reversed.

Characteristics of included articles ...

9. p.9, l.27 – the second reference provided here should be no. 19 not 20 (?)

Author's comments: We have carefully reviewed all references to ensure that these were not mistakenly misaligned.

Discussion

10. 2nd paragraph – supply to clients. The focus of the review is on procurement (i.e. availability) of MA medicines which does not equal to appropriate access. This should be made explicit here or in the limitations paragraph. Even if medicines were available at clinics and pharmacies, the medical abortion service provision was likely curtailed due to lock downs and priority given to covid patients. Moreover, if only relying on pharmacies and drug shops or online orders, information on effective regimens provided to women might be poor (e.g., Footman et al. 2018 'Medical Abortion Provision by Pharmacies and Drug Sellers in Low- and Middle-Income Countries: A Systematic Review).

Author's comments: This is an important clarification. We have removed "to clients" in the discussion and added a line in the strengths and limitations highlighting that availability does not infer to access to clients.

Strengths and limitations

11. l.11 'an overview' instead of 'on overview'

Author's comment: Corrected.

Conclusion

12. SMOs – please spell out

Author's comment: Corrected.

13. The role of social marketing organisations is highlighted (in abstract-conclusion and manuscript conclusion). The results section, however, does not make it entirely clear if the observations relate to public health systems, services provided by non-governmental organisations, or both. One mention of ministries suggests that the available evidence refers to public systems. It is important to clarify this point to provide an accurate account of what is known and what is not known. Supporting references for social marketing organisations distributing 'the bulk of medical abortion medicines' being distributed (and procured?) and their 'outsized influence' are needed.

Author's comment:

- *We noted the same lack of clarity as the documentation does not explicitly describe to whom or at what level the observations relate. (see Strengths and Limitations - p.13)*
- *Supporting reference with regards to SMO dominance of procurement and distribution added. Note that references are not typically found in conclusions sections.*

Reviewer: 2

Dr. Madeleine Ennis, The University of British Columbia Faculty of Medicine

Comments to the Author:

Summary:

The authors conducted a scoping review to systematically map the available literature on the disruptions to medical abortion medications caused by the COVID-19 pandemic, with two unpublished

articles included, demonstrating limited evidence of brief disruptions to medical abortion medications in Sub-Saharan Africa early in the pandemic. This publication makes a strong argument for needing to understand barriers to accessing medical abortion medications, and has the potential to facilitate collaboration, and allow researchers, medical staff, clinicians, and policy makers to stay up-to-date with issues in modern abortion care. It highlights that more research is needed to understand this issue, as there is a real scarcity in data. After a Pubmed search, I concluded that this is original research, with unique objectives.

This being said, I recommend substantial changes to this article.

Firstly, the language in the manuscript should be altered to reflect biological sex, and not gender (e.g. refer to females instead of women when making general statements). This language recognizes diversity and is more inclusive. The comment "Women in this scoping review is intended to be construed inclusively to include girls and all persons who can become pregnant." could then be removed.

Author's comment: We maintained the original language as this was what was used in the source material and follows WHO standard.

There is missing punctuation and a number of typos that should be corrected.

Author's comment: Thank you for this comment. We have read through the manuscript carefully made the corrections.

I would recommend a more in-depth description of medical abortion medications, and the pre-pandemic availability of these medications around the world, and especially in sub-Saharan Africa.

Author's comments: A description of the mechanism of action of mifepristone and misoprostol has been added, page 6, line 6-9. We have also added proxy information regarding availability of MA medicines based on the EML and where there are grounds for legal abortion, page 7, line 21-25.

Many of the conclusions and statements made throughout the article need to be revised to reflect that the available data was focused on sub-Saharan Africa and was unpublished, so as to not over state or over generalize their findings.

Author's comments: In one instance, "select sub-Saharan African countries" was specified to qualify the statement.

Specific comments:

Introduction:

1. Be consistent abbreviating MA.

Author's comments: Medical abortion is used in the abstract. After the initial written out form, MA is used in the rest of manuscript.

2. The introduction should include a more information on the number of abortions accessed annually using medications. What are the currently available medications, and how does their

availability differ around the world prior to the pandemic? Not just from a disruptions stand-point, but also from a regulations and legality perspective.

Author's comments: Annual abortions and % using medication abortion method was added along with a basic description of the medications page 15, line 12-19. We have found no reliable figure regarding the availability of abortion medications in Sub Saharan Africa pre or during the pandemic. This is of course in part due to stigma throughout the value chain. The lack of transparency is part of the problem

Methods:

1. The limitation that the studies refer to SRH including MA should be made clearer in the methods.

Author's comments: We have added this limitation to the strengths and limitations section after the abstract, as per Reviewer 1's suggestions.

Review findings:

2. It is very unclear throughout this section which medications are being described. Mifepristone, misoprostol, methotrexate, etc. Please elaborate on this throughout.

Author's comments: Descriptions of misoprostol and mifepristone have been included in the Introduction. There was no specific mention of either misoprostol or mifepristone in the context of procurement in the articles referenced.

Discussion:

1. You make it clear that there is a lack of literature on this topic and more research is needed to address the issue of MA medication access. However, your discussion is too short, and does not give the readers and good understanding of the global landscape of abortion care access, including access to medications. There should be more comparison to the literature, and authors thoughts on further solutions to this problem, possibly by describing facilitators to other medications around the world.

Author's comments: We agree that it would have been helpful to provide a brief snapshot of the global landscape of abortion care access and we have done our best to use proxy information to fill in gaps, page 15. We did seek to include data relevant to MA procurement such as estimates of medical abortion by region or even globally, and breakdowns of abortion by type of procedure, yet were unable to find any reliable and recent statistics. We originally included oxytocin as a search term in our scoping review but this did not yield any articles related to abortion during the pandemic period.

2. The limitation that the articles found focus on sub-Saharan Africa should be discussed more thoroughly throughout the discussion, including providing more context as to how abortion care is normally accessed there.

Author's comments: The first paragraphs of the discussion addresses these concerns.

Small details:

1. Consider switching medicines to medications, and be consistent in pluralizing it throughout.

Author's Comments: We have used medicines as per WHO preference.

Reviewer: 1

Competing interests of Reviewer: No competing interests.

Reviewer: 2

Competing interests of Reviewer: N/A

VERSION 2 – REVIEW

REVIEWER	Ennis, Madeleine The University of British Columbia Faculty of Medicine
REVIEW RETURNED	29-Sep-2022

GENERAL COMMENTS	Summary: Thank you for the opportunity to review this manuscript again. It is interesting and improved since its initial submission. Minor comments can be found below. Specific comments: Abstract: 1. Introduce the abbreviation MA in the Objective section2. The words “largely addressed” in the Findings are overstating your findings. Please rephrase. Introduction: 1. Do not need to produce MA multiple times in the introduction Discussion: 1. Please provide a reference for: “8.2 million annual abortions occurring annually in Africa, 76% are unsafe.” Also, remove one “annual” as this is stated twice. Small details: 1. Consider switching medicines to medications, and be consistent in pluralizing it throughout.
---

VERSION 2 – AUTHOR RESPONSE

Reviewer: 2

Dr. Madeleine Ennis, The University of British Columbia Faculty of Medicine Comments to the Author:

Summary:

Thank you for the opportunity to review this manuscript again. It is interesting and improved since its initial submission. Minor comments can be found below.

Specific comments:

Abstract:

1. Introduce the abbreviation MA in the Objective section

Author’s comments: This abbreviation has been added. Please reference page 2, l.4.

2. The words “largely addressed” in the Findings are overstating your findings. Please rephrase.

Author’s comments: The words “largely addressed” changed to “alleviated”. Please see page 2, l.27.

Introduction:

1. Do not need to produce MA multiple times in the introduction

Author’s comments: Noted. Please see page 4, l.10

Discussion:

1. Please provide a reference for: “8.2 million annual abortions occurring annually in Africa, 76% are unsafe.” Also, remove one “annual” as this is stated twice.

Author’s comments: The references were included, and the repetition removed. Please see page 11, l.28-29.

Small details:

1. Consider switching medicines to medications, and be consistent in pluralizing it throughout.
Author's comments: Respectfully, we prefer using the term medicines as this is what is commonly used by WHO. We trust that this is acceptable.